

# The avifauna of Ramanathapuram, Tamil Nadu along the Southeast coast of India: waterbird assessments and conservation implications across key sanctuaries and Ramsar sites

Hameed Byju[1], Hegde Maitreyi[2], Raveendran Natarajan[2], Reshmi Vijayan[3] and Balu Alagar Venmathi Maran[4]

[1] Centre of Advanced Study in Marine Biology, Annamalai University, Parangipettai, Tamil Nadu, India
[2] Iragukal Amritha Nature Trust, Thirumangalam, Madurai, Tamil Nadu, India
[3] Department of Zoology, BJM Government College, Kollam, Kerala, India
[4] Graduate School of Integrated Science and Technology, Nagasaki University, Nagasaki, Japan

Corresponding author
Balu Alagar Venmathi Maran,
bavmaran@nagasaki-u.ac.jp

## ABSTRACT

**Background**. Wetlands, globally, face significant threats from human activities, and waterbirds, as key indicators of wetland health, are essential to maintaining ecological balance. Any long-term conservation measures should prioritize coordinated habitat preservation, wetland restoration, and sustainable management practices involving local communities. Monitoring and analyzing waterbird population trends are critical for understanding restoration, conservation, and management practices.

**Methods**. The present study was carried out in five bird sanctuaries Chitrangudi, Kanjirankulam (Ramsar sites), Therthangal, Sakkarakottai, and Mel-Kel Selvanoor of Tamil Nadu, Southeast coast of India, over one year (April 2022 to March 2023). Monthly surveys using direct and block methods, with additional fortnightly visits during the breeding season, were conducted from vantage points to record species diversity, nesting activity, and conservation threats. Assessments of the residential status, national status (SOIB), and Convention for Migratory species (CMS) status were done along with the alpha and beta biodiversity profiles, principal component analysis, Pearson correlation and other statistical methods performed to assess breeding waterbirds community structure. Threats to the breeding waterbirds were categorised into high, medium, and low impacts based on degree of severity and irreversibility.

**Results**. The avifaunal checklist revealed a diversity of waterbird species utilizing the sanctuaries for breeding. Notable findings include two Near-Threatened species like, Asian Woolly-necked Stork *Ciconia episcopus,* and Spot-billed Pelican *Pelecanus philippensis,* where Asian Woolly-necked Stork recorded only in Therthangal Bird Sanctuary. Avifauna of each sanctuary with breeding waterbirds in parenthesis is as follows: Chitragundi 122 (13); Mel-Kel Selvanoor 117 (19); Therthangal 96 (23); Sakkarakottai 116 (17) and Kanjirankulam 123 (14). The breeding activity (incubation in nests) was from November to February except for Glossy Ibis and Oriental Darter whose breeding started in December; Spot-billed Duck and Knob-billed Duck breed only during January and February. Among the 131 species recorded from all the sanctuaries, 78% were resident birds; 27% were breeding waterbirds, and 21% were

Winter visitors. The SOIB and CMS statuses underscore the necessity of implementing effective conservation measures to protect breeding habitats amid anthropogenic pressures. Water unavailability and nest tree unavailability in the sanctuaries are found to be the high degree threats to breeding waterbirds than others. This research provides critical baseline data for the forest department's future wetland management plans.

# INTRODUCTION

Wetlands serve as crucial waterbird habitats, playing vital roles in feeding and breeding habitats; stop over sites and wintering grounds for migratory birds (*Chapman et al., 2001*; *Piersma & Lindström, 2004*; *Grimmett & Inskipp, 2007*; *Anand et al., 2023*). Any degradation of these habitats leads to a drop in the water table, disruptions in the food chain leading to declines in breeding and migratory waterbird populations (*Urfi, Sen & Megnathan, 2005*), posing detrimental impacts on the environment, ecosystems, and human well-being (*Kumar & Kankaujia, 2014*). Therefore, understanding the composition of bird communities is essential for identifying suitable local landscapes for avifaunal conservation and associated ecosystems (*Kattan & Franco, 2004*). Waterbirds play a crucial role in the nutrient cycles of wetlands, spanning various trophic levels, and act as bio-indicators (*Canterbury et al., 2000*; *Urfi, Sen & Megnathan, 2005*; *Sekercioglu, 2012*). Wetlands are highly productive ecosystems (*Paracuellos, 2006*) and various elements influence the composition of avifauna in wetland ecosystems (*Rajpar & Zakaria, 2010*), encompassing factors like wetland size (*Paracuellos, 2006*), water depth, duration, seasonal fluctuations (*Lagos et al., 2008*), water quality (*Hoyer & Caneld, 1994*). Factors like human-induced disturbances, habitat loss, habitat alterations negatively influence the waterbird diversity and abundance (*Craig & Barclay, 1992*; *Chawaka et al., 2017*; *Golzar et al., 2019*; *Halassi et al., 2022*). Approximately 64% of the world's wetlands have been lost since 1900, with the South Asian region experiencing even greater declines. Inland wetlands are vanishing more rapidly than coastal wetlands (*Ramsar Fact Sheet, 2014*). Numerous wetlands in India, including ones in Tamil Nadu whose 7% geographical area is wetlands, face the risk of degradation and loss due to expanding developmental and commercial activities (*SAC, 2011*; *Fraser & Keddy, 2005*). Lakes, reservoirs, rivers, tanks, and ponds are the main forms of wetland areas in Tamil Nadu state (*SAC, 2011*). Small tanks primarily built to store water for domestic consumption and irrigation (*Subramanya, 2005*), provide excellent feeding and nesting sites for colonial nesting birds.

On the Southeast coast of India, Point Calimere and Gulf of Mannar are two Important Bird and Biodiversity Areas (IBAs) in Tamil Nadu forming important stopover sites of Central Asian Flyway (CAF) for long distance migratory birds, along with Chitrangudi and Kanjirankulam bird sanctuaries which are Ramsar sites (*Islam & Rahmani, 2004*; *Rashiba et al., 2022*). Tamil Nadu state has the highest number of Ramsar sites ($n = 16$,

20% of total Ramsar sites in India) (*Tamil Nadu Wetland Mission, 2024*). Among 25 sanctuaries in the state, 15 are bird sanctuaries, among which 11 are heronries (Chitrangudi, Kanjirankulam, Karaivetti, Koonthankulam, Melselvanoor–Kelselvanoor, Udayamarthandpuram, Vaduvoor, Vellode, Vedanthangal, Karikili, and Vettangudi). Thirunelveli, Ramanthapuram, and Kancheepuram districts have the maximum numbers of heronries in Tamil Nadu, followed by Madurai, Chennai, and Nilgiris districts (*Subramanya, 2005*). Various avifaunal diversity studies on waterbirds have been done in Tamil Nadu, in Tiruppur (*Priya & Varunprasath, 2018*); central Tamil Nadu (*Krishnaraj & Mathesh, 2023*); Pallikaranai, Chennai (*Raj et al., 2010*); Karaivetti, Ariyalur (*Gokula, 2010*); Vaduvoor, Tiruvarur (*Gokula & Raj, 2011*); Samanatham, Madurai (*Byju et al., 2023a*). *Guptha et al. (2011)* studied 69 wetlands across eight districts of Tamil Nadu, where they recorded 53 wetland species.

In Ramanathapuram district, avian studies have historically focused on coastal birds (*Ali & Ripley, 1987*; *Balachandran, 1990*). Recent studies from the district include the avifaunal distribution of islands of the Gulf of Mannar (*Byju, Raveendran & Ravichandran, 2023b*), Valinokkam (*Byju et al., 2023c*), Karangadu mangroves (*Byju et al., 2023d*) and Melselvanoor–Kelselvanoor Bird Sanctuary (MKBS) (*Byju, Raveendran & Ravichandran, 2023e*). These one-year study sites are the two Ramsar sites, Kanjirankulam Bird Sanctuary (KBS) and Chitrangudi Bird Sanctuary (CBS), MKBS, Sakkarakottai Bird Sanctuary (SBS), and Therthangal Bird Sanctuary (TBS). Despite having the maximum number of bird sanctuaries in the Ramanathapuram region, there is no available literature on comprehensive study that assessed the avifaunal diversity, and seasonal patterns. There is a significant gap in understanding the waterbirds, and the breeding populations from these sanctuaries due to insufficient data hindering the development of effective conservation strategies tailored to the specific ecological needs of the waterbirds in the region as all these rainfed sanctuaries faced tree wilting due to anthropogenic stress, removal of water from the tanks for agricultural purposes affecting the waterbird population. To address this research gap, we pursued the following objectives. (i) To assess and compare the diversity and composition of bird species across the five sanctuaries; their residential status, Convention on the conservation of migratory species (CMS) status and State of India's Birds (SOIB) status; (ii) To investigate the breeding diversity of waterbirds and document the breeding season, abundance, and nesting preferences of abundant waterbird species from all sanctuaries; (iii) To assess the relationship between the area of the sanctuaries and diversity and richness of breeding waterbirds, testing the hypothesis that larger areas support high diversity and richness; (iv) To assess various threats to the breeding waterbirds and contribute towards the management plan of the forest department for conservation measures of the wetlands and its breeding waterbird species.

## MATERIALS & METHODS

### Study area

The study area included five bird sanctuaries from Tamil Nadu on the southeast coast of India. They are (i) Chitrangudi Bird Sanctuary (CBS) (9°19′N and 78°28′E) has

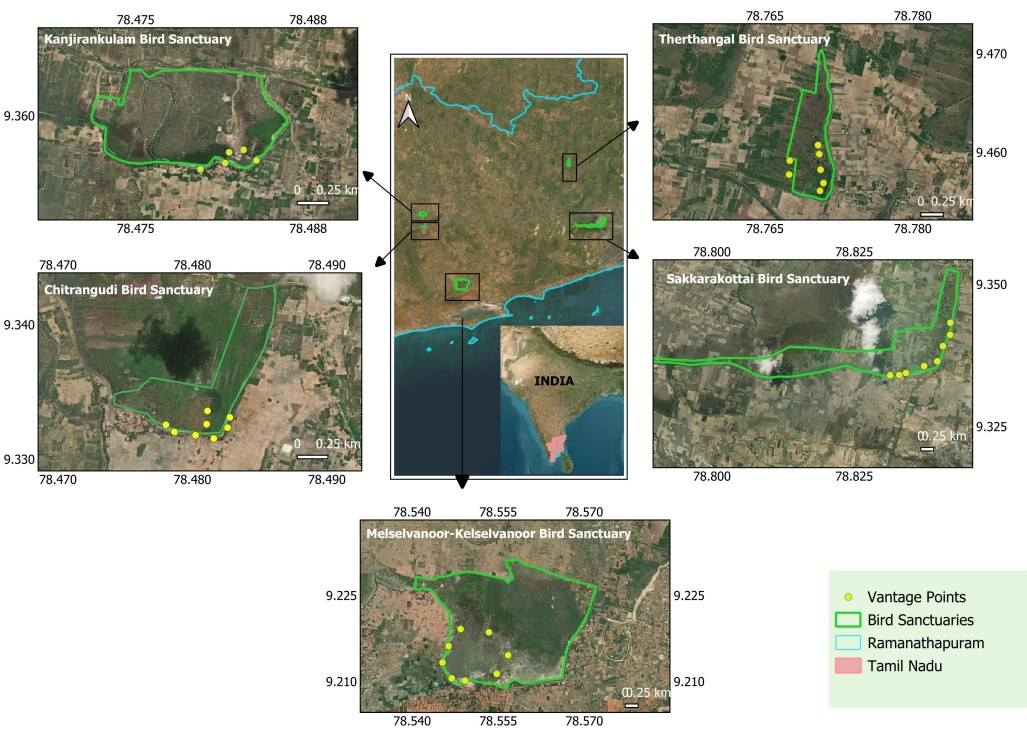

**Figure 1  Study areas and vantage points in all the five bird sanctuaries.** Study areas and vantage points in all the five bird sanctuaries of Ramanathapuram (map created using QGIS software with basemap ©ESRI Satellite).

an area of 47.63 ha. (ii) Melselvanoor Kelaselvanoor Bird Sanctuary (MKBS) (9°13′–9°12′N and 78°32′–78°34′E) has a total ayacut area of 429.15 ha. (iii) Kanjirankulam Birds Sanctuary (KBS) (9°21′N and 78°30′E) has an area of 98 ha. (iv) Sakkarakottai Birds Sanctuary (SBS) (9°21′N and 78°48′E) has an estimated area of 230.49 ha. (v) Therthangal Bird Sanctuary (TBS) (9°27′N and 78°46′E) an area of 29.295 ha (Fig. 1). All the sanctuaries are seasonal water holding community tanks with one meter to five meters depth. Elevation of all sanctuaries ranges between 30 m to 100 m mean sea level. Between the tank embankments and the vegetation, there is approximately a 15–100 m wide water holding region. Agricultural lands surround the sanctuaries. The prominent tree, *Acacia nilotica* (Babul) offers conducive breeding and feeding grounds for the waterbirds. Besides this, trees like *Prosopis juliflora*, *Tamarindus indica*, *Ficus* spp., *Thespesia populnea*, *Albizzia amara,* and Palmyra *(Borassus flabellifer)* are also found (*Byju & Raveendran, 2023a*; *Byju & Raveendran, 2023b*; *Byju & Raveendran, 2024a*; *Byju & Raveendran, 2024b*; *Byju et al., 2024a*). The main habitat types observed in the sanctuaries included: Open-water habitat, agricultural land, trees like *Babul,* mesquite *Prosopis juliflora*, palmyra *Borassus flabellifer,* and tamarind *Tamarindus indica* trees on the bund bordering the wetland, grassland on the wetland area, and shrub habitat.

Three distinct seasons are experienced in all the sanctuary areas. The winter (November–February); summer (March–June); scanty rains (July–September); and monsoon (October–December) as monsoon and winter seasons overlap with an average rainfall of 350 mm to 900 mm annually. Most of the water collected in the tanks is from the Northeast monsoon. Ramanathapuram Wildlife Division gave verbal permission to conduct the work.

## Data collection

A study on avifaunal diversity was conducted between April 2022 and March 2023. Twelve field visits (one per month) to assess the bird diversity, residential status, and breeding activity. Surveys were conducted in the morning (07.00 h–10.00 h) and the evening (16.00 h–19.00 h), during peak bird activity, following direct count and block count methods (*Bibby et al., 2000*; *Howes & Bakewell, 1989*). Three experienced observers and two field assistants conducted counts at vantage points: eight at CBS, five at KBS, seven at TBS, eight at SBS, and eight at MKBS separated by 100–200 m, depending on the landscape and visibility. Birds were counted for 15 min at each point, after a five-minute acclimation period. The observations recorded while moving from one scanning point to another were entered as incidental records. Birds were observed using Nikon binoculars ($10 \times 50$) and photographed using a Canon 100–400 mm lens. Breeding activity of waterbirds was documented through fortnightly surveys, during the breeding season (the time the birds occupied the nests during incubation). Identification of active nests was achieved through monitoring of flights of adult birds between nests and feeding grounds. The number of nests was estimated employing standard methods such as ground counts or nest counts using binoculars and spotting scopes (*Gibbs et al., 1988*; *Dodd & Murphy, 1995*). Other methods were impractical for sites with few nests or those completely inaccessible, therefore we used perimeter counts based on visible nests and observed foraging flights from the colony boundary (*Dodd & Murphy, 1995*). The common name, scientific name, IUCN Red List status, and migratory status are followed (*Praveen & Jayapal, 2023*). Species residential status as Resident (R), Passage Migrant (PM), or Winter Visitor (WV) depending on the temporal patterns and duration of occurrence (*Grimmett et al., 2011*) while global Red List status (*IUCN, 2024*), Convention of the Conservation of Migratory Species of Wild Animals (CMS) protection status (*The Convention on Migratory Species, 2024*) and national bird abundance trends (*State of India's Birds, 2023*) were assessed for the recorded species. According to SOIB, the current population trend of the bird species in India is the average annual change in the species abundance over the past eight years (2015–2022). Different categories of current population trend indices are, Insufficient Data means too few reports, Trend Inconclusive means 95% confidence interval >2%, Rapid Decline is decline >2.7%, Decline is >1.1%, Increase is >0.9% and Rapid Increase is >1.6% (*State of India's Birds, 2023*). Potential threats to birds based on impact of threat index across sanctuaries were noted for management recommendations (*Borgman, 2011*).

## Data processing

Alpha diversity Indices like Shannon-Weiner Diversity Index (H), Pielou's Evenness Index, Menhinick's Index for richness and Dominance Index (D) which implies presence of

one or few dominating species in the site, were performed to understand the overall community structure of breeding waterbirds in each sanctuary (*Kitikidou et al., 2024*). Whittaker's Beta-diversity Index measures the similarity in diversity between the study sites (*Whittaker, 1972*). Kruskal–Wallis test was performed to understand whether there was a statistically significant difference in the abundance of breeding waterbirds among the five sanctuaries (*Kruskal & Wallis, 1952*). The correlation between sanctuary area and the breeding waterbirds diversity was confirmed by Pearson correlation analysis. Further, principal component analysis (PCA) was performed to assess whether some waterbird families were more strongly associated with specific sanctuaries, using abundances of each waterbird family recorded in the sanctuaries as a variable (*Jolliffe, 2002*). Bray–Curtis cluster analysis, and individual rarefaction analysis were done for breeding waterbirds to understand the similarity in diversity and species accumulation among the sanctuaries and overall comprehensive community structure for each bird sanctuary (*Gotelli & Colwell, 2001*). All the analysis was done using Paleontological Statistics (PAST) Software version 4.17. Heatmap of species abundnace and other graphs were plotted using R version 4.4.1. Threats to breeding waterbirds were analysed for their extent of impact based on severity score, scope score and irreversibility score of the threat categories. Severity was assessed based on potential impacts of the threats or degrees of damage to the species; scope score was assessed based on the geographical extent to which the threat affects the landscape, whether the threat impacts the entire sanctuary, half of the sanctuary or create localized effect on a small portion of the sanctuary; based on whether the effects of these threats can be reversed or they create a permanent damage, irreversibility score was assessed. Four-scale measurements were used for all these scores as follows: 4 = extremely high, 3 = high, 2 = medium, and 1 = low. Total was calculated using the formula: Total = 2*(severity + scope) + irreversibility (*Margoluis & Salafsky, 1998*; *Rajashekara & Venkatesha, 2017*). Based on the results, threats were classified as High, Medium, and Low categories.

## RESULTS

### Avifauna

We recorded 131 species of birds compiled from all the sanctuaries together. The maximum number of bird species was recorded from KBS (123) followed by CBS (122), MKBS (117), SBS (116), and TBS (96). Based on the residential status, Resident birds (R) were predominant with 78% ($n = 102$) of the total species recorded whereas, Winter Visitors (WV) contributed 21% ($n = 28$). One species, Rosy Starling *Pastor roseus*, a Passage Migrant (PM) was common to all the sanctuaries. Table 1 summarizes the avifaunal list (order & family wise) compiled from all five sanctuaries with residential status and IUCN Red List categories. Twenty-seven bird species in the sanctuaries are protected under the CMS (Table 2). The avifaunal species listed in Appendix II of CMS correspond to migratory species that need international cooperation and international agreements for conservation and management (*The Convention on Migratory Species, 2024*). The national trends of the avifauna recorded from all sanctuaries, 40.4% of species populations are stable; rapid increase (4.5%); increase (4.5%); decline (12.9%); rapid decline (11.4%); trend inconclusive (23.6%) as per assessment by *State of India's Birds (2023)* (Table 3).

**Table 1** Avifaunal data depicting species, families, residents, migrants and IUCN status from the five sanctuaries.

|  | CBS | KBS | MKBS | SBS | TBS |
|---|---|---|---|---|---|
| **Total species** | 122 | 123 | 117 | 116 | 96 |
| **Number of orders** | 19 | 19 | 19 | 19 | 18 |
| **Number of families** | 53 | 53 | 52 | 52 | 45 |
| **% of Residents** (n = number of species) | 80 (n = 98) | 73 (n = 98) | 83 (n = 98) | 83 (n = 96) | 86 (n = 83) |
| **% of Winter migrants** (n = number of species) | 19 (n = 23) | 19 (n = 24) | 15 (n = 18) | 16 (n = 19) | 12 (n = 12) |
| **Least concerned species**[*] | 120 | 121 | 115 | 114 | 94 |
| **Near threatened species**[*] | 1 | 1 | 1 | 1 | 2 |
| **Vulnerable species**[*] | 1 | 1 | 1 | 1 | 0 |

**Notes.**

CBS, Chitrangudi Bird Sanctuary; KBS, Kanjirankulam Birds Sanctuary; MKBS, Melselvanoor Kelaselvanoor Bird Sanctuary; SBS, Sakkarakottai Bird Sanctuary; TBS, Therthangal Bird Sanctuary.

*Classified according to IUCN Red List category.

Except for some species, most bird species observed are common to all the five sanctuaries. Passeriformes order was predominant with 22 families and 41 species in CBS, MKBS, KBS, and SBS, except for TBS which is only represented by 18 families and 29 species. Family Ardeidae predominated in all the sanctuaries (represented by 11 species in CBS, MKBS, and KBS; 10 in SBS and TBS respectively). According to the IUCN Red List status, two Near Threatened species namely, Asian Woolly-necked Stork *Ciconia episcopus* (only recorded in TBS) and, Spot-billed Pelican *Pelecanus philippensis* and one Vulnerable species Indian Spotted Eagle *Clanga hastata* are commonly found in all the five sanctuaries. Migratory species such as Northern Shoveler *Spatula clypeata*, Baillon's Crake *Zapornia pusilla*, and resident birds like Watercock *Gallicrex cinerea*, were recorded only in TBS. Similarly, in KBS, we recorded the winter visitor Green Sandpiper *Tringa ocropus* which was not recorded from other sanctuaries. Table 3 summarizes the avifaunal list of all sanctuaries depicting the presence or absence of species with SOIB trends.

## Breeding waterbird diversity

We recorded 55 waterbird species recorded from all sanctuaries. Out of which 51% of species (n = 28) are breeding in these wetlands. Grey Heron *Ardea cinerea*, Black-crowned Night Heron *Nycticorax nycticorax*, Purple Heron *Ardea purpurea*, Asian Openbill *Aanastomus oscitans*, Black-headed Ibis, and Little Cormorant *Microcarbo niger* were found to breed in all the five sites. Five species namely Common Moorhen *Gallinula chloropus*, Pheasant-tailed Jacana *Hydrophasianus chirurgus*, Grey Headed Swamphen *Porphyrio poliocephalus*, White-breasted Waterhen *Amaurornis phoenicurus,* and Little Grebe *Tachybaptus ruficollis* were least common, found to breed only in SBS (Fig. 2). Except for SBS, Asian Openbill nested predominantly in the other four sanctuaries followed by Painted Stork, Black-headed Ibis, and Spot-billed Pelican. It was observed in all the sanctuaries that Spot-billed Pelicans and Painted Storks occupy the large *Acacia nilotica* trees which are dominantly cultivated as fuel wood for nearby villagers in all the tanks. Trees like *Prosopis juliflora* are occupied by Asian Openbill while Black-headed Ibis occupy the canopy. Cormorants, egrets, and pond

**Table 2  Convention For Migratory Species of Wild Animals.** Bird Species which are protected under Convention For Migratory Species of Wild Animals (CMS).

| S. No | Species | CMS status |
|---|---|---|
| 1 | Purple Heron | Appendix II |
| 2 | Eurasian Wigeon | Appendix II |
| 3 | Common Teal | Appendix II |
| 4 | Northern Pintail | Appendix II |
| 5 | Knob-billed Duck | Appendix II |
| 6 | Garganey | Appendix II |
| 7 | Northern Shoveler | Appendix II |
| 8 | Eurasian Coot | Appendix II |
| 9 | Baillon's Crake | Appendix II |
| 10 | Common Redshank | Appendix II |
| 11 | Common Snipe | Appendix II |
| 12 | Wood Sandpiper | Appendix II |
| 13 | Common Sandpiper | Appendix II |
| 14 | Marsh Sandpiper | Appendix II |
| 15 | Green Sandpiper | Appendix II |
| 16 | Eurasian Spoonbill | Appendix II |
| 17 | Glossy Ibis | Appendix II |
| 18 | Great Egret | Appendix II |
| 19 | Little Stint | Appendix II |
| 20 | Temminck's Stint | Appendix II |
| 21 | Black-winged Stilt | Appendix II |
| 22 | Common Kestrel | Appendix II |
| 23 | Black Kite | Appendix II |
| 24 | Booted Eagle | Appendix II |
| 25 | Shikra | Appendix II |
| 26 | Oriental Honey Buzzard | Appendix II |
| 27 | Western Marsh Harrier | Appendix II |

herons occupy the lower strata of the habitat. Abundance of all breeding waterbird species in each sanctuary in the breeding months is represented in Fig. 3. The overall trend in the monthly abundance of breeding waterbirds in each sanctuary showed that in TBS, highest abundance was recorded in November and December, whereas in SBS it was in January and February. In MKBS, CBS and KBS, the abundance increased slightly from October to November and remained almost similar until February (Fig. 4).

## Chitrangudi Bird Sanctuary

CBS supports 42 species of waterbirds and 13 species are recorded to be breeding. The waterbirds breeding season (incubating in nests) started from November 2022 to February 2023. For two species, Glossy Ibis and Oriental Darter, the breeding season started in December 2022.

**Table 3  Checklist of all the five bird sanctuaries.** Avifaunal checklist of all the five bird sanctuaries with migration, IUCN and SOIB status.

| Scientific name | Common name | Migration status | IUCN status | Species recorded | | | | | Population trend in India (*State of India's Birds, 2023*) |
|---|---|---|---|---|---|---|---|---|---|
| | | | | CBS | KBS | MKBS | TBS | SBS | |
| **Order: Galliformes** | | | | | | | | | |
| **Family: Phasianidae** | | | | | | | | | |
| *Pavo cristatus* | Indian Peafowl | R | LC | * | * | * | * | * | Rapid Increase |
| *Francolinus pondicerianus* | Grey Francolin | R | LC | * | * | * | * | * | Increase |
| **Order: Anseriformes** | | | | | | | | | |
| **Family: Anatidae** | | | | | | | | | |
| *Anas Penelope* | Eurasian Wigeon | WV | LC | * | * | - | - | * | Decline |
| *Anas crecca* | Common Teal | WV | LC | * | * | * | - | * | Rapid Decline |
| *Anas acuta* | Northern Pintail | WV | LC | * | * | * | - | * | Rapid Decline |
| *Sarkidiornis melanotos* | Knob-billed Duck | R | LC | * | * | * | * | * | Trend Inconclusive |
| *Anas poecilorhyncha* | Indian Spot-billed Duck | R | LC | * | * | * | * | * | Stable |
| *Spatula querquedula* | Garganey | WV | LC | * | * | * | * | * | Rapid Decline |
| *Anser indicus* | Bar-headed Goose | WV | LC | * | * | - | - | - | Decline |
| *Dendrocygna javanica* | Lesser Whistling Duck | R | LC | - | - | - | * | * | Stable |
| *Spatula clypeata* | Northern Shoveler | WV | LC | - | - | - | * | - | Rapid Decline |
| **Order: Podicipediformes** | | | | | | | | | |
| **Family: Podicipedidae** | | | | | | | | | |
| *Tachybaptus ruficollis* | Little Grebe | R | LC | * | * | * | * | * | Trend Inconclusive |
| **Order: Piciformes** | | | | | | | | | |
| **Family: Picidae** | | | | | | | | | |
| *Dinopium benghalense* | Black-rumped Flameback | R | LC | * | * | * | * | * | Trend Inconclusive |
| **Family:Megalaimidae** | | | | | | | | | |
| *Psilopogon haemacephalus* | Coppersmith Barbet | R | LC | * | * | * | * | * | Stable |
| **Order: Bucerotiformes** | | | | | | | | | |
| **Family: Upupidae** | | | | | | | | | |
| *Upupa epops* | Common Hoopoe | R | LC | * | * | * | * | * | Trend Inconclusive |
| **Order: Coraciiformes** | | | | | | | | | |
| **Family: Coraciidae** | | | | | | | | | |
| *Coracias benghalensis* | Indian Roller | R | LC | * | * | * | * | * | Decline |
| **Family: Alcedinidae** | | | | | | | | | |
| *Ceryle rudis* | Pied Kingfisher | R | LC | * | * | * | - | * | Decline |
| *Halcyon smyrnensis* | White-throated Kingfisher | R | LC | * | * | * | * | * | Trend Inconclusive |
| *Alcedo atthis* | Common Kingfisher | R | LC | * | * | * | * | * | Trend Inconclusive |
| *Halcyon pileata* | Black-capped Kingfisher | WV | LC | * | * | * | - | - | Rapid Decline |
| **Family: Meropidae** | | | | | | | | | |
| *Merops orientalis* | Green Bee-eater | R | LC | * | * | * | * | * | Stable |
| *Merops philippinus* | Blue-tailed Bee- eater | WV | LC | * | * | * | * | * | Rapid Increase |
| **Order: Cuculiformes** | | | | | | | | | |

**Table 3** (*continued*)

| Scientific name | Common name | Migration status | IUCN status | Species recorded | | | | | Population trend in India (*State of India's Birds, 2023*) |
|---|---|---|---|---|---|---|---|---|---|
| | | | | CBS | KBS | MKBS | TBS | SBS | |
| **Family: Cuculidae** | | | | | | | | | |
| *Centropus sinensis* | Greater Coucal | R | LC | * | * | * | * | * | Rapid Increase |
| *Eudynamys scolopaceus* | Asian Koel | R | LC | * | * | * | * | * | Increase |
| *Clamator jacobinus* | Pied Cuckoo | R | LC | * | * | * | * | * | Stable |
| *Hierrococcyx varius* | Common Hawk Cuckoo | R | LC | * | * | * | * | * | Rapid Increase |
| **Order: Psittaciformes** | | | | | | | | | |
| **Family: Psittacidae** | | | | | | | | | |
| *Psittacula krameri* | Rose-ringed Parakeet | R | LC | * | * | * | * | * | Trend Inconclusive |
| **Order: Strigiformes** | | | | | | | | | |
| **Family: Strigidae** | | | | | | | | | |
| *Athene brama* | Spotted Owlet | R | LC | * | * | * | * | * | Not Available |
| **Order: Columbiformes** | | | | | | | | | |
| **Family: Columbidae** | | | | | | | | | |
| *Columba livia* | Rock Pigeon | R | LC | * | * | * | * | * | Increase |
| *Streptopelia decaocto* | Eurasian Collared-Dove | R | LC | * | * | * | * | * | Increase |
| *Spilopelia senegalensis* | Laughing Dove | R | LC | * | * | * | * | * | Trend Inconclusive |
| *Spilopelia chinensis* | Spotted Dove | R | LC | * | * | * | * | * | Increase |
| **Order: Apodiformes** | | | | | | | | | |
| **Family: Apodidae** | | | | | | | | | |
| *Cypsiurus balasiensis* | Asian Palm- swift | R | LC | * | * | * | * | * | Insufficient Data |
| *Apus melba* | Alpine Swift | R | LC | * | * | * | - | * | Stable |
| **Order: Gruiformes** | | | | | | | | | |
| **Family: Rallidae** | | | | | | | | | |
| *Gallinula chloropus* | Eurasian Moorhen | R | LC | * | * | * | - | * | Stable |
| *Porphyrio Poliocephalus* | Grey-headed Swamphen | R | LC | * | * | * | * | * | Stable |
| *Amaurornis phoenicurus* | White-breasted Waterhen | R | LC | * | * | * | * | * | Trend Inconclusive |
| *Fulica atra* | Eurasian Coot | R | LC | * | * | * | * | * | Decline |
| *Zapornia pusilla* | Baillon's Crake | WV | LC | - | - | - | * | - | Rapid Decline |
| *Gallicrex cinerea* | Watercock | R | LC | - | - | - | * | - | Stable |
| **Order: Charadriiformes** | | | | | | | | | |
| **Family: Rostratulidae** | | | | | | | | | |
| *Rostratula benghalensis* | Greater Painted Snipe | WV | LC | * | * | - | - | - | Stable |
| **Family: Scolopacidae** | | | | | | | | | |
| *Tringa totanus* | Common Redshank | WV | LC | * | * | - | - | * | Decline |
| *Gallinago gallinago* | Common Snipe | WV | LC | * | * | - | - | * | Trend Inconclusive |
| *Tringa glareola* | Wood Sandpiper | WV | LC | * | * | * | * | * | Decline |
| *Actitis hypoleucos* | Common Sandpiper | WV | LC | * | * | * | * | * | Decline |
| *Tringa stagnatilis* | Marsh Sandpiper | WV | LC | * | - | - | - | * | Rapid Decline |
| *Tringa ocropus* | Green Sandpiper | WV | LC | - | * | - | - | - | Stable |
| *Calidris minuta* | Little Stint | WV | LC | - | * | * | - | - | Rapid Decline |
| *Calidris temminickii* | Temminck's Stint | WV | LC | - | * | * | - | - | Trend Inconclusive |

(*continued on next page*)

**Table 3** (*continued*)

| Scientific name | Common name | Migration status | IUCN status | Species recorded | | | | | Population trend in India (*State of India's Birds, 2023*) |
|---|---|---|---|---|---|---|---|---|---|
| | | | | CBS | KBS | MKBS | TBS | SBS | |
| **Family: Jacanidae** | | | | | | | | | |
| *Hydrophasianus chirurgus* | Pheasant-tailed Jacana | R | LC | * | * | * | * | * | Decline |
| **Family: Burhinidae** | | | | | | | | | |
| *Burhinus indicus* | Indian Stone-curlew | R | LC | * | * | * | - | * | Insufficient data |
| **Family: Charadriidae** | | | | | | | | | |
| *Charadrius dubius* | Little Ringed Plover | WV | LC | * | * | * | - | * | Rapid Decline |
| *Vanellus malabaricus* | Yellow-wattled Lapwing | R | LC | * | * | * | - | * | Stable |
| *Vanellus indicus* | Red-wattled Lapwing | R | LC | * | * | * | * | * | Trend Inconclusive |
| **Family: Recurvirostridae** | | | | | | | | | |
| *Himantopus himantopus* | Black-winged Stilt | R | LC | * | * | * | * | * | Trend Inconclusive |
| **Family: Laridae** | | | | | | | | | |
| *Chlidonias hybrida* | Whiskered Tern | WV | LC | * | * | * | * | * | Rapid Decline |
| **Order: Falconiformes** | | | | | | | | | |
| **Family: Falconidae** | | | | | | | | | |
| *Falco tinnunculus* | Common Kestrel | WV | LC | * | * | * | - | * | Rapid Decline |
| **Order: Accipitriformes** | | | | | | | | | |
| **Family: Accipitridae** | | | | | | | | | |
| *Milvus migrans* | Black Kite | R | LC | * | * | * | - | * | Trend Inconclusive |
| *Elanus caeruleus* | Black-winged Kite | R | LC | * | * | * | * | * | Decline |
| *Hieraaetus pennatus* | Booted Eagle | WV | LC | * | * | * | * | * | Trend Inconclusive |
| *Accipiter badius* | Shikra | R | LC | * | * | * | * | * | Stable |
| *Pernis ptilorhynchus* | Oriental Honey Buzzard | R | LC | * | * | * | * | * | Stable |
| *Clanga hastata* | Indian Spotted Eagle | R | VU | * | * | * | | * | Trend Inconclusive |
| *Haliastur indus* | Brahminy Kite | R | LC | * | * | * | * | * | Stable |
| *Circus aeruginosus* | Western Marsh Harrier | WV | LC | - | - | - | * | - | Decline |
| **Order: Suliformes** | | | | | | | | | |
| **Family: Anhingidae** | | | | | | | | | |
| *Anhinga melanogaster* | Oriental Darter | R | LC | * | * | * | * | * | Stable |
| **Family: Phalacrocoracidae** | | | | | | | | | |
| *Microcarbo niger* | Little Cormorant | R | LC | * | * | * | * | * | Stable |
| *Phalacrocorax carbo* | Great Cormorant | R | LC | * | * | * | * | * | Stable |
| *Phalacrocorax fuscicollis* | Indian Cormorant | R | LC | * | * | * | * | * | Trend Inconclusive |
| **Order: Pelicaniformes** | | | | | | | | | |
| **Family: Ardeidae** | | | | | | | | | |
| *Lxobrychus sinensis* | Yellow Bittern | R | LC | * | * | * | * | * | Trend Inconclusive |
| *Ardea cinerea* | Grey Heron | R | LC | * | * | * | * | * | Trend Inconclusive |
| *Ardea purpurea* | Purple Heron | R | LC | * | * | * | * | * | Stable |
| *Egretta garzetta* | Little Egret | R | LC | * | * | * | * | * | Trend Inconclusive |
| *Bubulcus ibis* | Cattle Egret | R | LC | * | * | * | * | * | Stable |
| *Ardea alba* | Great Egret | R | LC | * | * | * | * | * | Trend Inconclusive |
| *Ardea intermedia* | Intermediate Egret | R | LC | * | * | * | * | * | Trend Inconclusive |
| *Ardeola grayii* | Indian Pond Heron | R | LC | * | * | * | * | * | Stable |
| *Nycticorax nycticorax* | Black-crowned Night Heron | R | LC | * | * | * | * | * | Stable |

| Scientific name | Common name | Migration status | IUCN status | Species recorded | | | | | Population trend in India (*State of India's Birds, 2023*) |
|---|---|---|---|---|---|---|---|---|---|
| | | | | CBS | KBS | MKBS | TBS | SBS | |
| *Butorides striata* | Striated Heron | R | LC | * | * | * | * | * | Trend Inconclusive |
| *Egretta gularis* | Western Reef Heron | R | LC | * | * | * | - | - | Decline |
| **Family: Threskiornithidae** | | | | | | | | | |
| *Threskiornis melanocephalus* | Black-headed Ibis | R | LC | * | * | * | * | * | Stable |
| *Plegadis falcinellus* | Glossy Ibis | R | LC | * | * | * | * | * | Stable |
| *Pseudibis papillosa* | Red-naped Ibis | R | LC | * | * | * | * | * | Stable |
| *Platalea leucorodia* | Eurasian Spoonbill | R | LC | * | * | * | * | - | Rapid Decline |
| **Family: Pelecanidae** | | | | | | | | | |
| *Pelecanus philippensis* | Spot-billed Pelican | R | NT | * | * | * | * | * | Rapid Decline |
| **Order: Ciconiiformes** | | | | | | | | | |
| **Family: Ciconiidae** | | | | | | | | | |
| *Anastomus oscitans* | Asian Openbill | R | LC | * | * | * | * | * | Trend Inconclusive |
| *Mycteria leucocephala* | Painted Stork | R | LC | * | * | * | * | * | Decline |
| *Ciconia episcopus* | Asian Woolly-necked Stork | R | NT | - | - | - | * | - | Decline |
| **Order: Passeriformes** | | | | | | | | | |
| **Family: Campephagidae** | | | | | | | | | |
| *Coracina macei* | Large Cuckooshrike | R | LC | * | * | * | - | * | Stable |
| **Family: Oriolidae** | | | | | | | | | |
| *Oriolus kundoo* | Indian Golden Oriole | WV | LC | * | * | * | - | * | Stable |
| **Family: Artamidae** | | | | | | | | | |
| *Artamus fuscus* | Ashy Wood Swallow | R | LC | * | * | * | * | * | Stable |
| **Family: Vanghidae** | | | | | | | | | |
| *Tephrodornis pondicerianus* | Common Woodshrike | R | LC | * | * | * | - | * | Stable |
| **Family: Laniidae** | | | | | | | | | |
| *Lanius vittatus* | Bay-backed Shrike | R | LC | * | * | * | - | * | Stable |
| *Lanius cristatus* | Brown Shrike | WV | LC | * | * | * | * | * | Stable |
| *Lanius schach* | Long-tailed Shrike | | | - | - | - | * | - | Stable |
| **Family: Dicruridae** | | | | | | | | | |
| *Dicrurus macrocercus* | Black Drongo | R | LC | * | * | * | * | * | Stable |
| **Family: Monarchidae** | | | | | | | | | |
| *Terpsiphone paradisi* | Indian Paradise-flycatcher | R | LC | * | * | * | - | * | Stable |
| **Family: Corvidae** | | | | | | | | | |
| *Dendrocitta vagabunda* | Rufous Treepie | R | LC | * | * | * | * | * | Stable |
| *Corvus macrorhynchos* | Indian Jungle Crow | R | LC | * | * | * | * | * | Stable |
| *Corvus splendens* | House Crow | R | LC | * | * | * | * | * | Trend Inconclusive |
| **Family: Sturnidae** | | | | | | | | | |
| *Acridotheres tristis* | Common Myna | R | LC | * | * | * | * | * | Stable |
| *Pastor roseus* | Rosy Starling | PM | LC | * | * | * | * | * | Rapid Decline |
| *Sturnia pagodarum* | Brahminy Starling | R | LC | * | * | * | - | * | Trend Inconclusive |

| Scientific name | Common name | Migration status | IUCN status | Species recorded | | | | | Population trend in India (*State of India's Birds, 2023*) |
|---|---|---|---|---|---|---|---|---|---|
| | | | | CBS | KBS | MKBS | TBS | SBS | |
| **Family: Hirundinidae** | | | | | | | | | |
| *Cecropis daurica* | Red-rumped Swallow | R | LC | * | * | * | - | * | Stable |
| *Hirundo rustica* | Barn Swallow | WV | LC | * | * | * | * | * | Decline |
| **Family: Pycnonotidae** | | | | | | | | | |
| *Pycnonotus cafer* | Red-vented Bulbul | R | LC | * | * | * | * | * | Trend Inconclusive |
| **Family: Timaliidae** | | | | | | | | | |
| *Turdoides affinis* | Yellow-billed Babbler | R | LC | * | * | * | * | * | Stable |
| *Turdoides malcolmi* | Large Grey Babbler | R | LC | * | * | * | - | * | Stable |
| **Family: Cisticolidae** | | | | | | | | | |
| *Prinia socialis* | Ashy Prinia | R | LC | * | * | * | * | * | Increase |
| *Prinia inornata* | Plain Prinia | R | LC | * | * | * | * | * | Stable |
| *Orthotomus sutorius* | Common Tailorbird | R | LC | * | * | * | * | * | Rapid Increase |
| *Cisticola juncidis* | Zitting Cisticola | R | LC | * | * | * | * | * | Stable |
| **Family: Acrocephalidae** | | | | | | | | | |
| *Acrocephalus dumetorum* | Blyth's Reed Warbler | WV | LC | * | * | * | * | * | Stable |
| **Family: Alaudidae** | | | | | | | | | |
| *Eremopterix griseus* | Ashy-crowned Sparrow Lark | R | LC | * | * | * | - | * | Trend Inconclusive |
| *Galerida cristata* | Jerdon's Bushlark | R | LC | * | * | * | * | * | Stable |
| *Alauda gulgula* | Oriental Skylark | R | LC | * | * | * | - | * | Rapid Decline |
| **Family: Muscicapidae** | | | | | | | | | |
| *Muscicapa dauurica* | Asian Brown Flycatcher | WV | LC | * | * | * | - | * | Stable |
| *Copsychus fulicatus* | Indian Robin | R | LC | * | * | * | * | * | Stable |
| *Copsychus saularis* | Oriental Magpie Robin | R | LC | * | * | * | * | * | Stable |
| *Saxicola caprata* | Pied Bushchat | R | LC | - | - | - | * | - | Stable |
| **Family: Nectariniidae** | | | | | | | | | |
| *Cinnyris lotenius* | Loten's Sunbird | R | LC | * | * | * | - | * | Rapid Increase |
| *Cinnyris asiaticus* | Purple-rumped Sunbird | R | LC | * | * | * | * | * | Trend Inconclusive |
| *Cinnyris asiaticus* | Purple Sunbird | R | LC | * | * | * | * | * | Trend Inconclusive |
| **Family: Ploceidae** | | | | | | | | | |
| *Ploceus philippinus* | Baya Weaver | R | LC | * | * | * | * | * | Stable |
| **Family: Estrildidae** | | | | | | | | | |
| *Euodice malabarica* | Indian Silverbill | R | LC | * | * | * | * | * | Trend Inconclusive |
| *Lonchura malacca* | Tricoloured Munia | R | LC | * | * | * | - | * | Stable |
| **Family: Dicaeidae** | | | | | | | | | |
| *Dicaeum concolor* | Pale-billed Flowerpecker | R | LC | * | * | * | * | * | Stable |
| **Family: Passeridae** | | | | | | | | | |
| *Passer domesticus* | House Sparrow | R | LC | * | * | * | * | * | Decline |
| *Gymnoris xanthocollis* | Yellow-throated Sparrow | R | LC | * | * | * | - | * | Stable |
| **Family: Motacillidae** | | | | | | | | | |
| *Motacilla maderaspatensis* | White-browed Wagtail | R | LC | * | * | * | * | * | Stable |
| *Anthus rufulus* | Paddyfield Pipit | R | LC | * | * | * | * | * | Decline |

**Notes.**
*Presence.
⁻Absence.
R, Resident; WV, Winter visitor; PM, Passage migrant; CR, Critically Endangered; EN, Endangered; LC, Least Concern; NT, Near Threatened; VU, Vulnerable.

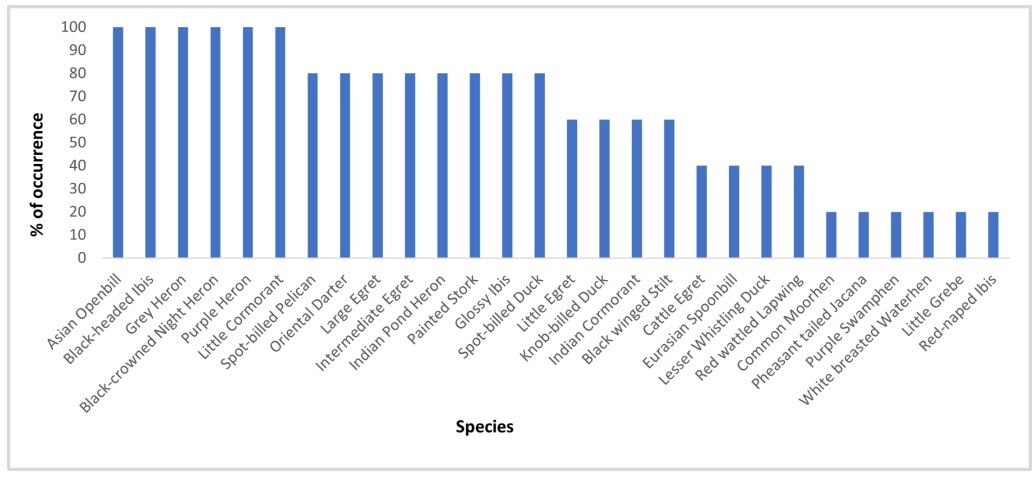

**Figure 2  Breeding waterbirds in sanctuaries.** Percentage of occurrence of breeding waterbirds in all the sanctuaries.

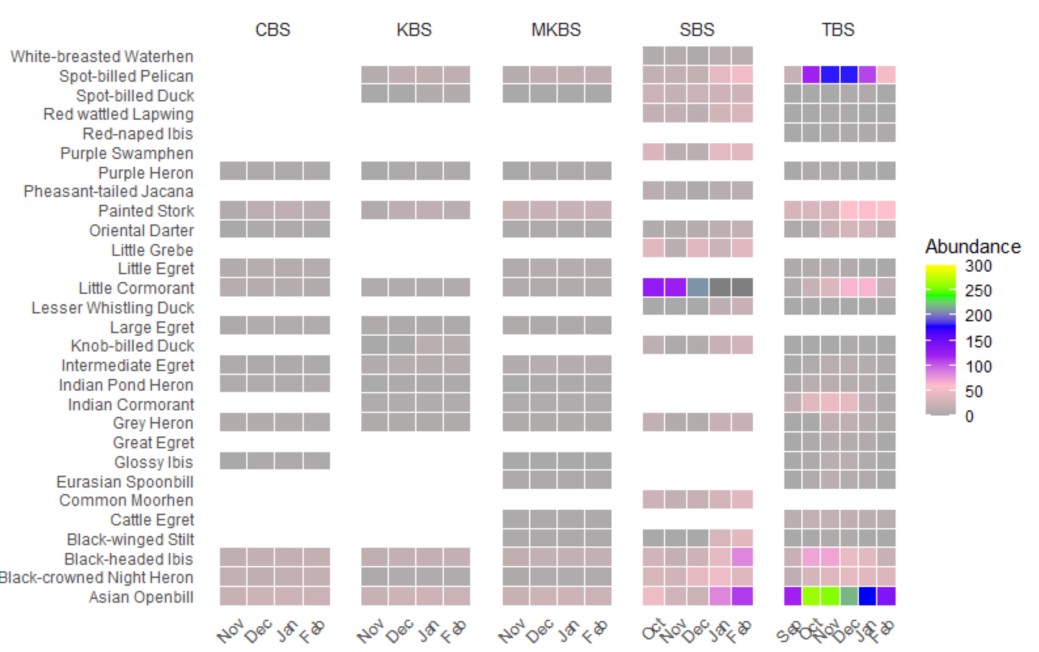

**Figure 3  Abundance of waterbirds.** Monthly abundance of breeding waterbirds in each sanctuary.

## Kanjirankulam Bird Sanctuary

In KBS, 49 species of waterbirds were recorded and 14 species are breeding. For all the species breeding season starts from November 2022 and extends until February 2023 except for Spot-billed Duck and Knob-billed Duck which breeds during January and February 2023. Interestingly, Oriental Darter, Cattle Egret *Bubulcus ibis*, Little Egret, Glossy Ibis *Plegadis falcinellus*, Eurasian Spoonbill and Black-winged Stilt *Himantopus himantopus* which were found to nest at the nearby sanctuaries were not reported from here.
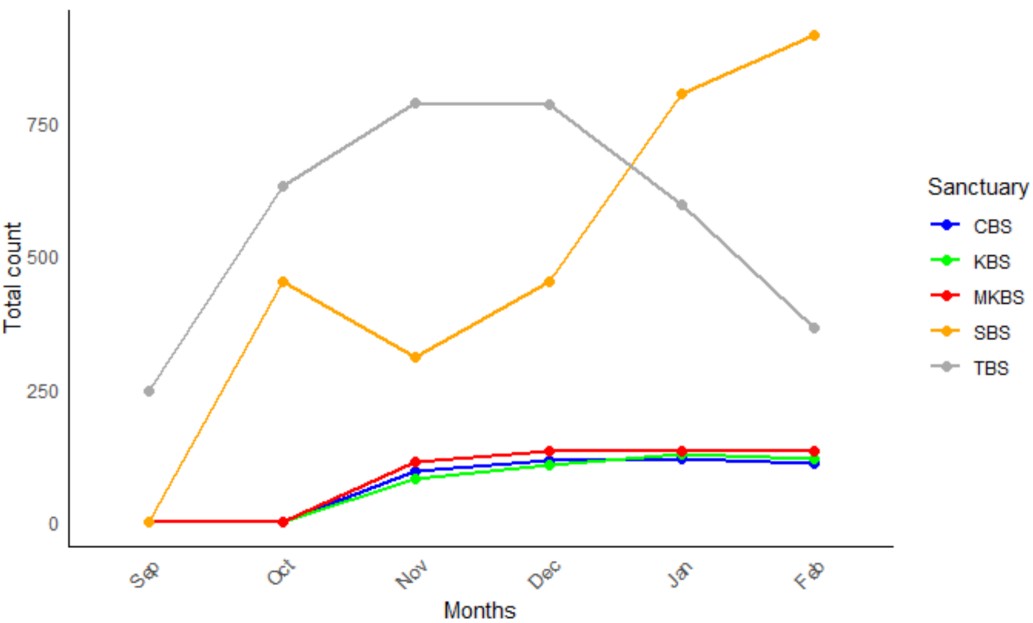

**Figure 4  Monthly abundance of breeding waterbirds.** Trend in monthly abundance of breeding waterbirds across five sanctuaries.

### Melselvanoor–Kelselvanoor Bird Sanctuary

MKBS provided a breeding habitat for 19 waterbird species. The breeding season starts in November 2022 and extends until February 2023 for all the species with no exception. Four species—Oriental Darter, Painted Stork, Black-headed Ibis, and Spot-billed Pelican—nest together in this sanctuary.

### Sakkarakottai Bird Sanctuary

SBS recorded 43 species of waterbirds and 17 species breed here. The breeding season in this sanctuary is from October 2022 to February 2023.

### Therthangal Bird Sanctuary

TBS recorded 40 species of waterbirds and 23 breeds here. The breeding season (birds incubating in nests) started in September 2022 and extended until February 2023. Three species, like Black-headed Ibis, Oriental Darter, and Spot-bellied Pelican, nest together in this sanctuary. Red-naped Ibis is found to breed only in TBS.

### Community structure of breeding waterbirds in the five sanctuaries

The Shannon-Weiner Index (H) revealed the highest breeding waterbird diversity at MKBS (2.54) followed by KBS (2.39), SBS (2.3), CBS (2.29), and the lowest at TBS (2.23). Dominance indices indicated balanced communities in CBS, KBS, and MKBS, while SBS and TBS, were dominated by a few species like Asian Open-bill, Painted Stork and Spot-billed Pelican. The Evenness Index revealed high evenness in CBS (0.7) and KBS (0.7), followed by MKBS (0.6), SBS (0.5), and TBS (0.4). Menhinick's Index ranked MKBS highest in species richness (0.8), followed by CBS (0.6) and KBS (0.6), with SBS and

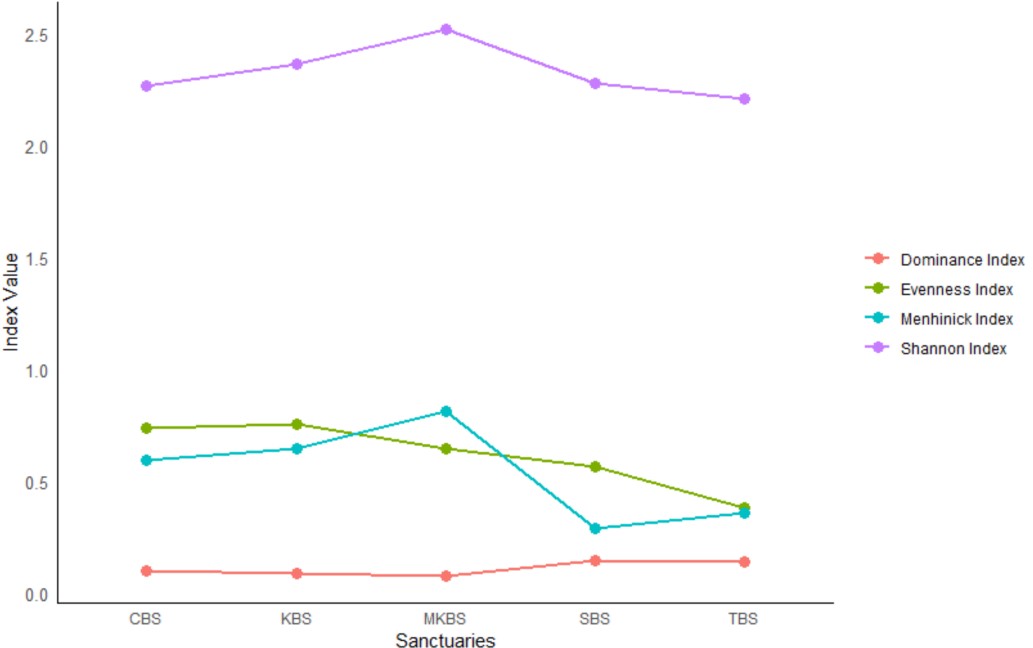

**Figure 5 Diversity indices for waterbirds.** Alpha diversity indices for breeding waterbirds in the sanctuaries.

**Table 4 Whittaker beta diversity profile of the sanctuaries in pairs.**

|  | CBS | KBS | MKBS | SBS | TBS |
|---|---|---|---|---|---|
| **CBS** | 0 | 0.25926 | 0.1875 | 0.6 | 0.27778 |
| **KBS** | 0.25926 | 0 | 0.21212 | 0.48387 | 0.24324 |
| **MKBS** | 0.1875 | 0.21212 | 0 | 0.5 | 0.095238 |
| **SBS** | 0.6 | 0.48387 | 0.5 | 0 | 0.4 |
| **TBS** | 0.27778 | 0.24324 | 0.095238 | 0.4 | 0 |

TBS showing the lowest richness (0.3) (Fig. 5 and Table S1). Overall, MKBS exhibited the highest diversity, and species richness with low dominance, whereas, TBS showed the lowest diversity, richness, and evenness dominated by a few species. These indices together provide a comprehensive view of the community structure for each bird sanctuary. Whittaker's beta diversity profile revealed that CBS and KBS are highly similar to MKBS. MKBS and TBS showed high similarity in diversity with each other. SBS showed distinctness in diversity and more similarity to TBS than to other sanctuaries (Table 4). In general, lower values in Whittaker's beta-diversity index indicate high similarity between the sites. Kruskal–Wallis test showed that there is no statistically significant difference ($p = 0.406 > 0.05$) in breeding waterbird diversity and richness among the sanctuaries. Further, the positive correlation between the area of the sanctuaries to the breeding waterbird diversity ($r = 0.8$, $p = 0.074$) and richness ($r = 0.4$, $p = 0.424$) was confirmed by Pearson correlation analysis although the correlation was not statistically significant.

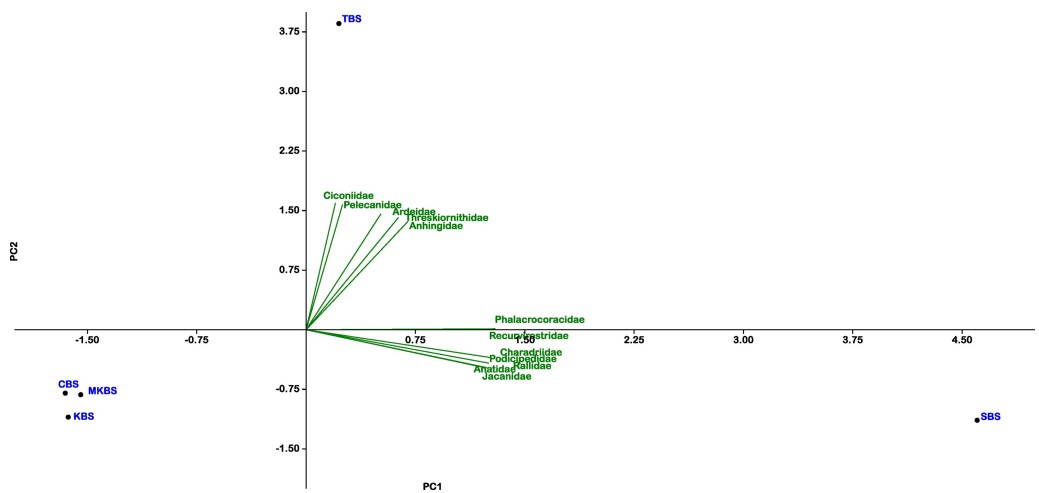

**Figure 6  Principal component analysis plot of various breeding waterbird families and sanctuaries.**

**Table 5  Principal component analysis (Eigenvalues and variance values).**

| PC | Eigenvalue | % variance |
|---|---|---|
| 1 | 7.25489 | 60.457 |
| 2 | 4.66775 | 38.898 |
| 3 | 0.06002 | 0.50017 |
| 4 | 0.017339 | 0.14449 |

The PCA analysis showed the association of families like Ciconiidae, Pelecanidae, Anhingidae, Ardeidae, Threskiornithidae specifically within TBS, whereas families Jacanidae, Anatidae, Rallidae, Podicipedidae, Phalacrocoracidae and Recurvirostridae more associated with SBS and comparatively less to TBS. CBS, MKBS and KBS did not show specific associations with any particular families and showed a negative correlation with other families (Fig. 6). Eigenvalues of the components are given in Table 5. Rarefaction curves (Fig. 7) indicated TBS had the highest curve, (20 species, 3,500 individuals) suggesting further sampling may uncover more species. MKBS had 19 taxa, and SBS, though lower in richness (17 taxa), appeared sufficiently sampled. KBS and CBS showed the lowest species richness with only 13 and 14 taxa respectively, and fewer than 500 individuals. Bray–Curtis cluster analysis (Fig. 8) showed SBS and TBS shared the most similar species, while KBS and MKBS formed a closely related cluster. CBS is closely related to the cluster of KBS and MKBS. The largest dissimilarity is between the SBS and TBS and the combined cluster of CBS, KBS, and MKBS.

## Potential threats

The present study recorded potential threats to the wetlands due to anthropogenic activities. Threats were categorized to assess the severity on the breeding waterbirds in all the sanctuaries. Water unavailability and nesting tree unavailability were the high ranked threats to the breeding waterbirds. Firewood collection, recreation and other human

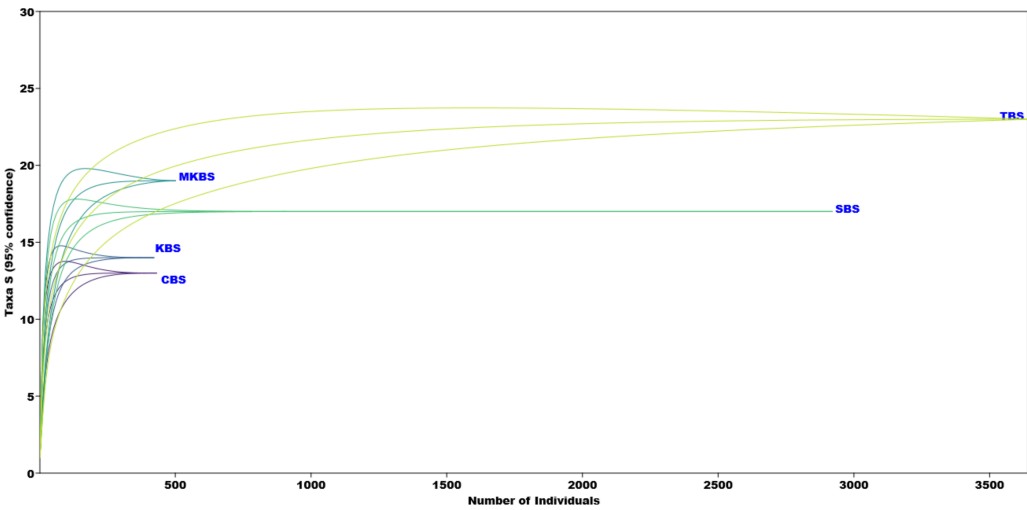

**Figure 7  Individual rarefaction curve of the five sanctuaries.**

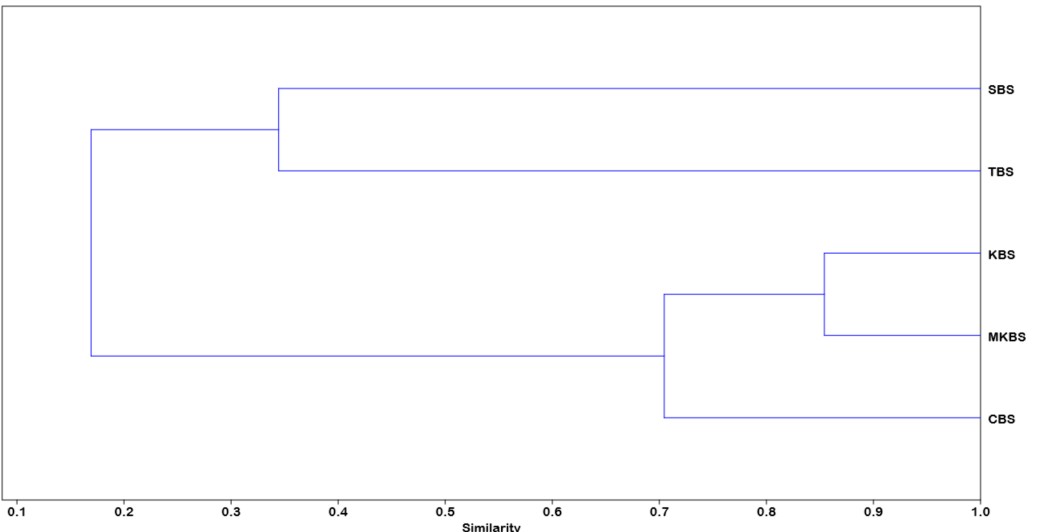

**Figure 8  Cluster analysis of the sanctuaries based on Bray–Curtis similarity index.**

disturbances were the medium category threats whereas livestock grazing, presence of invasive species, overfishing and feral dogs were the low impact threats, considering the severity, extent of impact and degree of irreversibility of the damage (Table S2).

## DISCUSSION

This study provided a comprehensive assessment of the avifaunal diversity across five bird sanctuaries, emphasizing the importance of these wetlands in supporting a rich diversity of bird species, particularly waterbirds. The study recorded 131 bird species across five sanctuaries, with the highest species richness observed in MKBS, followed by CBS, KBS,

SBS, and TBS. These wetlands play a crucial role in supporting diverse bird populations as evident form our results, particularly waterbirds with 51% being wetland-dependent and 21% winter visitors highlighting their biodiversity importance (*BirdLife International, 2017*; *Ma et al., 2010*). Passeriformes dominated in CBS, MKBS, KBS, and SBS, consistent with global patterns as the most diverse avian order (*Grimmett et al., 2011*). Ardeidae was the most common family, indicating the suitability of these wetlands for herons and egrets (*Byju, Raveendran & Ravichandran, 2023e*). The presence of Near-Threatened species Spot-billed Pelican across all sanctuaries along with the Vulnerable Indian Spotted Eagle, and one more Near-Threatened Asian Woolly-necked Stork in TBS emphasizes the conservation value of these habitats (*IUCN, 2024*). TBS was unique in recording migratory species like the Northern Shoveler and Bailon's Crake, as well as resident species like the Watercock highlighting the need for tailored conservation strategies for each sanctuary, acknowledging their unique contributions to regional biodiversity (*Mukherjee, Borad & Parasharya, 2002*). The study recorded notable breeding colonies of Asian Openbill, Grey Heron, and Black-crowned Night Heron. The preference of Spot-billed Pelicans and Painted Storks for native *Acacia nilotica* trees, and the stratified nesting behaviour observed, where different species occupy different canopy levels, reflect the ecological complexity of these habitats (*Subramanya, 1996*).

In earlier studies on heronries in Tamil Nadu, *Frank, Gopi & Pandav (2021)* found no nesting records in CBS, KBS, and SBS between 2017 and 2019 with MKBS and TBS not included in that study. *Byju, Raveendran & Ravichandran (2023e)* studied MKBS avifauna between 2016 and 2019 and recorded 115 bird species, including 19 breeding waterbirds. They observed Red-naped Ibis breeding at MKBS, which was not recorded breeding during the current study, while Black-winged Stilt which they did not observe breeding, was recorded as breeding which occurred from November to February in the present study (*Byju, Raveendran & Ravichandran, 2023e*). This year (2022–2023) our regular study with the forest department in CBS revealed significantly fewer nesting species and reduced diversity compared to the previous year's nesting populations. Regular breeders like Knob-billed Duck *Sarkidiornis melanotos*, Spot-billed Duck *Anas poecilorhyncha*, Eurasian Spoonbill *Platalea leucorodia,* and Spot-billed Pelican did not nest in CBS due to sparse rainfall and water scarcity. Similarly, some species breeding in nearby sanctuaries did not breed in KBS.

## Diversity and community structure across sanctuaries

Our hypothesis—that wetland area positively affects bird diversity and species richness—was broadly supported by the results, though nuances were evident in the distribution patterns among sanctuaries as MKBS, the largest sanctuary had the highest breeding waterbird diversity and the smallest sanctuary TBS had the lowest. These findings underscore the ecological importance of wetland heterogeneity, structural diversity, and habitat size in supporting complex waterbird communities. The Shannon-Weiner index ($H$) indicated that MKBS exhibited the highest diversity ($H = 2.54$) and suggesting a well-balanced avian community (*Magurran, 2004*), followed by KBS, SBS, CBS and TBS. Conversely, TBS showed the lowest diversity ($H = 2.23$) indicating a community structure

dominated by a few species, likely due to habitat degradation or limited resources (*Ludwig & Reynolds, 1988*) This is further reinforced by the dominance indices, which revealed balanced communities in CBS, KBS and MKBS, but community dominance by certain species in SBS and TBS. These patterns confirm that habitat complexity within larger wetlands likely fosters diverse avian assemblages through niche diversification. In terms of total area, MKBS and TBS are the largest and the smallest sanctuaries, respectively. It is evident by the correlation analysis that there is a strong positive relationship between sanctuary area and breeding waterbird diversity and richness. Larger the wetlands, more are the microhabitats and hence, they support greater bird diversity (*Paszkowski & Tonn, 2000*).

## Species richness and evenness among sanctuaries

Species richness, as quantified by Menhinick's Index, was highest at MKBS (0.8), with CBS and KBS following closely. SBS and TBS exhibited the lowest species richness values, indicating a restricted avian community in these areas, potentially due to smaller habitat size or fewer habitat types available to meet varied ecological needs. The Evenness Index, showing high evenness in CBS and KBS, further suggests a balanced distribution of species in these sanctuaries, where resources or niches may be more evenly distributed. Conversely, TBS, dominated by a few species, showed lower evenness, aligning with expectations that smaller or less varied wetlands may support more specialized assemblages where particular species outcompete others. These indices collectively indicate that MKBS's large area and CBS and KBS's habitat structures contribute positively to both species richness and evenness. Studies by *Fairbairn & Dinsmore (2001)* and *Riffel, Keas & Burton (2001)* demonstrated that wetland area and heterogeneity are the important features affecting bird richness.

## Beta diversity and inter-site community similarity

Whittaker's beta diversity profile highlights community similarities and distinctions among the sanctuaries. MKBS was most similar to CBS and KBS, with TBS and SBS showing distinct diversity profiles. SBS shared more similarities with TBS than other sanctuaries, suggesting that these smaller sites may host more unique assemblages. The high similarity observed between CBS, KBS, and MKBS could be attributed to shared habitat characteristics or landscape connectivity, which allows species to exploit resources across these areas. Lower values in Whittaker's index reflect this close similarity, suggesting possible species overlap or similar community dynamics facilitated by comparable habitat features.

## Relationship between wetland area, bird diversity, and richness

A Kruskal–Wallis test confirmed no statistically significant difference in waterbird diversity and richness across the sanctuaries ($p = 0.406$), suggesting that each sanctuary supports comparable community richness and diversity levels despite some ecological and structural variations. However, a positive Pearson correlation ($r = 0.8$, $p = 0.074$) between sanctuary area and waterbird diversity suggests that larger sanctuaries tend to support more diverse communities. While the correlation between area and species richness was positive ($r = 0.4$) but not statistically significant ($p = 0.424$), the results nonetheless suggest a

trend supporting the hypothesis that larger wetland areas tend to foster greater species richness.

## Species assemblage patterns and habitat associations

The PCA provided additional insights into species assemblage patterns and habitat associations across sanctuaries. TBS was associated with families such as Ciconiidae, Pelecanidae, Anhingidae, Ardeidae, and Threskiornithidae, which may reflect its unique ecological attributes or niche specialization supporting these groups. On the other hand, families like Jacanidae, Anatidae, Rallidae, Podicipedidae, Phalacrocoracidae, and Recurvirostridae were more strongly associated with SBS, suggesting habitat characteristics or resources better suited to these families. CBS, MKBS, and KBS did not show specific family associations, indicating that their habitats may provide a broader, less specialized environment, allowing for a more generalized assemblage without strong affiliations. This generalist habitat structure might contribute to the observed balanced community structure in these sanctuaries.

## Species status (CMS, SOIB)

As 20.6% of the species recorded from the study sites are enlisted in *The Convention on Migratory Species (2024)*, the significance of these sanctuaries for avifauna is critical for conservation management as this is the first study from all sanctuaries which could initiate international collaboration for species conservation. SOIB status assessment was considered for a better undertanding of the species distribution in all the five sanctuaries as there was no previous studies and published literature from these important sites for waterbirds. However, SOIB assessment considered the frequency of reporting to calculate the indices of abundance, which does not directly measure the population size. A 'complete checklist' in the e-Bird citizen science platform is considered to assess the frequency of reporting (*State of India's Birds, 2023*), which could be biased or contain errors due to bird detection and identification errors, inaccessibility of the birding locations due to seasonal changes or difficult terrains (*Maitreyi, 2024*). For the 'Insufficient Data' index, there are no measurable limits or range, hence this could be treated as a general trend and emphasize the value of our study due to data deficiency.

## Implications on conservation and management

Many migratory birds' flight paths depend on well-functioning wetlands (*Ens, Piersma & Tinbergen, 1994*; *Ntiamoa-Baidu et al., 1998*). Apart from resident and breeding birds, our study sites receive migratory birds majorly from families of Scolopacidae, Anatidae, Accipitridae, Charadriidae, Laridae, and Oriolidae. This remarkable diversity of avifauna in these study areas can be attributed to a wide range of feeding niches. The findings on threats have critical implications for conservation and habitat management in sanctuaries. As our results of threat ranking revealed, water unavailability is an extremely high threat in all the sanctuaries except TBS, where there is better availability of water as the area of TBS is much lesser. Another high ranked threat was nest tree unavailability which persists in all the sanctuaries. Although each of the five study sites has a large geographical area in water retention, the actual water holding area during the breeding season was much smaller due

to reliance on rainfall, which is also impacted by climate change and altered monsoon patterns, finally affecting colonial breeding waterbirds (*Cavitt et al., 2014*). As the water availability decreased, trees wilted, reducing the number of suitable nesting sites (personal observations). The primary roosting and nesting tree, *Acacia nilotica*, is also declining due to altered monsoon patterns, emphasizing the need for planting native species that are more resilient to drought and excess water conditions, to support waterbird breeding populations. Fluctuating water levels influence ecological factors like the size, depth, and vegetation affecting the availability of suitable breeding and roosting sites (*MacArthur, MacArthur & Preer, 1962*; *Karr & Roth, 1971*; *Pearman, 2002*) and shaping waterbird populations and species distribution in wetlands (*Wiens, 1989*; *Mukherjee, Borad & Parasharya, 2002*; *Ma et al., 2010*).The neighboring agricultural lands play a significant role in providing food resources for all these varieties of bird species (*Byju et al., 2024b*), especially in April when the water level in the waterbody recedes, allowing for the cultivation of crops like paddy *Oryza sativa*, cotton *Gossypium herbaceum*, and chilly *Capsicum annuum*, which attracts numerous birds. Therefore, conservation strategies should prioritize habitat preservation and structural diversification, especially in smaller wetlands like TBS and CBS, where specialized or less diverse habitats may restrict community diversity.

This study also observed the other medium and low category threats common to habitat and breeding waterbirds in all the sanctuaries, like proximity to human settlements and urban expansion (*Roshnath & Sashikumar, 2019*) which create disturbances as firewood collection, fishing or recreation by people, cattle feeding on the bark of the nesting trees during dry season which decayed the tree, leading to tree death (*Byju et al., 2024a*), domestic dogs disturbing and preying on waterbirds as observed in all the sanctuaries (*Mundkur & Langendoen, 2019*), invasive species like *Prosopis juliflora* and *Ipome a carnea* taking over a major extent of these sanctuaries and the removal of *Prosopis* trees without a replacement strategy, has negatively impacted species like the Asian Openbill and cormorants, which preferred nesting in these trees.. Sometimes waterbodies are also seen choked with water Hyacinth *Eichhornia crassipes*, which reduces feeding and roosting habitat for ducks and other waterbird species in freshwater that reduces oxygen and sunlight for native plants and fishes (*Lowe et al., 2000*; *Szabo & Mundkur, 2017*).

## Management recommendations

Human-wildlife conflicts over resources, such as fishing and water rights, further challenge bird conservation (*Bosselmann, Engel & Taylor, 2008*). To mitigate these threats, enhanced patrolling to limit overfishing, along with controlling firewood collection is essential, as dead trees provide crucial roosting sites. Collaboration with government agencies is needed to ensure perennial water in the sanctuaries, with occasional desilting and tank deepening to improve water-holding capacity (*Anand, 1999*). Raising public awareness and training local communities to monitor waterbird populations and threats can foster greater protection and conservation efforts (*Szabo et al., 2016*).

## CONCLUSIONS

To fully understand the ecological dynamics, seasonal bird abundance, diversity, and conservation in these five sanctuaries, long-term scientific studies, rigorous monitoring, and community involvement are essential. In summary, wetland size plays a critical role in shaping bird species richness and overall abundance, while the complexity of bird communities, indicated by species diversity, is more linked with habitat diversity, supporting the hypothesis that wetland area and habitat structure significantly influence avian community diversity, richness and evenness. Recent work on CAF reported a declining population trend of 100 waterbird species stressing the importance of forests and inland wetlands (*Mundkur & Selvaraj, 2023*). This research lays a foundation for future studies on wetland ecology, offering critical data for avian biodiversity conservation and sustainable wetland management across protected habitats.

## ACKNOWLEDGEMENTS

We are grateful to the Gulf of Mannar Wildlife Division and the Wildlife Warden for permitting us to conduct the fieldwork. Student volunteers, the forest staff, and the village communities also deserve a special mention for helping us during our fieldwork.

### Funding

The study was funded by the Integrated Development of Wildlife Habitats During 2023–2024 in Wildlife Division, Ramanathapuram. There was no other external funding received. The funders had no role in study design, data collection and analysis, decision to publish, or preparation of the manuscript.

### Grant Disclosures

The following grant information was disclosed by the authors:
The Integrated Development of Wildlife Habitats During 2023–2024 in Wildlife Division, Ramanathapuram.

### Competing Interests

Balu Alagar Venmathi Maran is an Academic Editor for PeerJ.

### Author Contributions

- Hameed Byju conceived and designed the experiments, performed the experiments, analyzed the data, prepared figures and/or tables, authored or reviewed drafts of the article, and approved the final draft.
- Hegde Maitreyi performed the experiments, analyzed the data, prepared figures and/or tables, and approved the final draft.
- Raveendran Natarajan analyzed the data, authored or reviewed drafts of the article, and approved the final draft.

- Reshmi Vijayan analyzed the data, authored or reviewed drafts of the article, and approved the final draft.
- Balu Alagar Venmathi Maran analyzed the data, authored or reviewed drafts of the article, and approved the final draft.

### Field Study Permissions

The following information was supplied relating to field study approvals (i.e., approving body and any reference numbers):

Gulf of Mannar Wildlife Division and the Wildlife Warden for permitting to conduct the fieldwork. Permission was given verbally.

### Data Availability

The raw data of water birds used for statistical analysis are available in the Supplementary File.

### Supplemental Information

Supplemental information for this article can be found online at http://dx.doi.org/10.7717/peerj.18899#supplemental-information.

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
