# Peer review of "The avifauna of Ramanathapuram, Tamil Nadu along the Southeast coast of India: waterbird assessments and conservation implications across key sanctuaries and Ramsar sites"

_PeerJ, doi:10.7717/peerj.18899_

## Round 0.1 · original submission · Major Revisions

Thank you very much for your manuscript titled “Avian Conservation Efforts in Tamil Nadu: Focus on Water Bird Breeding and Conservation from Five Bird Sanctuaries Along the Southeast Coast of India” that you sent to PeerJ.

This study provides valuable and important descriptive information on birds present in five sanctuaries in a region of India. The study is interesting in its capacity to record avifauna in systematically underexplored sites.

As you will see below, comments from two referees suggest a major revision before your paper can be published. The reviewers felt that several changes were needed throughout much of the manuscript, including the title, objectives, hypothesis, and methods. Their comments should provide a clear idea for you to review, hopefully improving the clarity and rigor of the presentation of your work. I will be happy to accept your article pending further revisions, detailed by the referees, which largely focus on clarifying various aspects of your work.

Reviewer 1 requests greater clarity in the objectives, in addition to fully complementing them with the methods used and the results obtained.

Reviewer 2 points out the absence of a hypothesis or research question, as well as improving the description of the methods used. It also considers it important to point out the absence of references that support the evidence reported, as well as complementing the results based on the reported methods.

Please note that we consider these revisions to be important and your revised manuscript will likely need to be revised again.

Reviewer 1 ·

Basic reporting

I am not a fluent speaker, and it is rather difficult for me to evaluate the English. But there are many sentences that I need help understanding.
Title. "Avian Conservation Efforts in Tamil Nadu: Focus on Water Bird Breeding and Conservation from Five Bird Sanctuaries Along the Southeast Coast of India"
I do not understand what it means to "Focus on Water Bird Breeding and Conservation."?
Abstract. Lines 21-46. I would advise re-writing the abstract after responding to all other comments.
I do not really understand how the authors define breeding and resident species. Needs clarification in the Abstract and throughout the manuscript.

The birds' national conservation status could also be taken into account. Authors might want to check the State of India's Birds (2023).
Also, they might want to check how many species in these sanctuaries are protected under the Ramsar Convention and Convention on Migratory Species.

Experimental design

Objective
Lines 106-109.
Objective (i) "To update the avifaunal distribution from all five bird sanctuaries including the two mentioned Ramsar sites" is not clear to me. Do the authors mean bird species composition or distribution of every bird species within each of the five sanctuaries?
Objective (iii), "To understand any bird-human negative interaction from all five sanctuaries based on the site's conservation significance," is also unclear. The bird-human negative interaction could be identified based on analysis of threats, human activities, or public survey but not based on the site's conservation significance. Also, in Objectives, the authors might want to state "what?" rather than "how?".
* * *
Methods
Lines 107-108 Objective "(ii) To estimate the population of breeding waterbirds..."
I could not find in the results any estimation of a population or a method that described the estimation of a population.
Lines 140-141 "active breeding nests"
Does it mean active nests?
Lines 147-149 "Birds were observed using Nikon binoculars (10x50) and photographed using Canon 100-400 mm lens and were identified later with the help of field guides"
Does it mean that the counters could not count the birds on site due to poor identification skills?
Lines 150-151 "The residential status of the birds was worked out based on the available literature and birds are grouped under different categories like Resident (R), Passage Migrant (PM), and Winter Visitor (WV) depending on their timing and duration of occurrence (Grimmett et al., 2011)".
What about species that only breed in the area and then migrate? Are there such species?
Do the authors decide on the breeding status based on a finding of occupied nests or also based on the secondary features (e.g., singing males, territorial defense, and so on)?
Line 152-153 "During the breeding season, the waterbird nesting counts were observed and are given separately".
What does it mean? Count of occupied nests?

Validity of the findings

There is a lot of work done on the collection of the data and its initial processing.
However, the objectives are not clear and there is a significant methodological gap between the declared objectives and obtained results.
Probably, with such material, the authors could focus on the assessment of the international conservation importance of the sanctuaries, on the evaluation of current and potential threats and existing conservation measures, and therefore can develop recommendations on improvement of conservation within the sanctuaries and beyond them at the policy level.

Additional comments

No additional comments.

Reviewer 2 ·

Basic reporting

The work presents 'Avian Conservation Efforts in Tamil Nadu: Focus on Water Bird Breeding and Conservation from Five Bird Sanctuaries Along the Southeast Coast of India.' I believe the title does not reflect the content, as it seems more of a descriptive characterization of the avifauna of five sites. The study is interesting in its capacity to record avifauna in systematically underexplored sites. However, I consider the work to be too descriptive, lacking a hypothesis grounded in a theoretical framework. Consequently, the results cannot test a hypothesis, and the scientific structure presented lacks the basic elements of scientific research. Additionally, no research question is posed to guide the work. I believe the figures presented could be improved, as they do not meet basic standards for publication. There are many claims without supporting evidence or relevant bibliographic references. I have left a series of comments inserted in the PDF file that I believe should be addressed to substantially improve the manuscript.

Experimental design

The study does not present a clear, objective research question with a description of its variables. Therefore, there is no connection to a theoretical framework within which the work can be framed, nor is there a corresponding hypothesis and guiding research question. The methods presented seem insufficient for reproducibility and replicability in science, particularly regarding the data analysis section. Some results are shown and discussed without a rigorous methodology for data collection. For example, results on human activities that constitute threats are presented, but no procedures are described in the methods for measuring these disturbances. I have left a series of comments inserted in the PDF file that I believe should be addressed to substantially improve the manuscript, thereby clarifying and transparently presenting the true results obtained from your methods.

Validity of the findings

The study mentions many findings but fails to measure or only weakly measures some aspects, such as threats or habitat characteristics, which were not detailed in the methods section. The analyses and conclusions do not honestly reflect what the authors have measured. The statistical analyses are more descriptive diversity indices, which could apply to a characterization of avifauna for a study site. It seems that solid statistical inference is not possible without predictor variables that the authors describe as measured, such as habitat characteristics and anthropogenic disturbances.

Additional comments

I believe the work presents many errors in scientific structure and methodology. The authors need to make a significant effort to address the major shortcomings highlighted in the attached PDF file. In my opinion, the title of the work does not reflect what is proposed in the manuscript and does not meet the minimum quality criteria of PeerJ, for example, it lacks a hypothesis and research question.

Annotated reviews are not available for download in order to protect the identity of reviewers who chose to remain anonymous.

---

## Round 0.2 · Major Revisions

Thank you very much for your manuscript titled “The Avifauna of Ramanathapuram, Tamil Nadu Along the Southeast Coast of India: Water Bird Assessments and Conservation Implications Across Key Sanctuaries and Ramsar Sites” that you sent to PeerJ.

This study presents very valuable and relevant information on diversity of waterbird species on a coast of India.

As you will see below, comments from referee suggests a major revision. Their comments should provide a clear idea for you to review, hopefully improving the clarity and rigor of the presentation of your work. There are still some very important points that need to be clarified. The introduction should expand on the theoretical framework that supports the work. It is also necessary to clarify what the hypothesis or research question of this study is. In the methods, it is necessary to clarify how the habitat analyses were carried out, the possible threats to wetlands and a greater emphasis on clarifying the statistical analyses used, in addition to improving the quality of the figures.

Please note that we consider these revisions to be important and your revised manuscript will likely need to be revised again.

Reviewer 2 ·

Basic reporting

The paper presents "The Avifauna of Ramanathapuram, Tamil Nadu Along the Southeast Coast of India: Water Bird Assessments and Conservation Implications Across Key Sanctuaries and Ramsar Sites." I believe the authors have made an effort to improve the manuscript, for instance, by updating and enhancing the references and incorporating some additional background information that adds value to the work. They have also adjusted the title to one that better reflects the results presented. However, it is important to emphasize that once again, there is no hypothesis or research question guiding the study. The work remains highly descriptive. While efforts have been made to improve Figure 1, the remaining figures (2–7) have not been improved to a basic professional standard required for scientific publication. I believe these latter aspects are crucial for the paper to meet the editorial scientific quality standards of the journal.

Experimental design

The study once again does not present a clear and objective research question, nor a description of its variables. As a result, there is no connection between a theoretical framework that could underpin the work, a corresponding hypothesis, and a guiding research question. While several methodological aspects I had previously suggested for improvement in the first version have been clarified, many issues remain unresolved. For example, how was the habitat analysis conducted? This is not detailed in the methods section, yet results are later presented, such as percentages of association with certain habitat types. However, no method is provided to measure this association, for instance, any model that would allow relating and determining the effect of certain habitats to infer possible bird use or occupation of specific habitats.

Validity of the findings

Although the findings are important from the perspective of wetland management and conservation, the study presents some results that are either not measured or weakly measured. For example, there is a section on potential threats that is not detailed in the methodology, yet it is presented as a result. This could have been discussed in the analysis section to interpret some results or something along those lines. Additionally, I believe a greater effort could have been made in the statistical analyses. Only rather descriptive diversity indicators are presented, which may be suitable for characterizing the avifauna of a study site, but I think it is not possible to draw solid statistical inferences without predictor variables, which the authors describe as being measured, such as habitat characteristics.

Additional comments

I suggest the authors consider submitting the manuscript to another journal with a different focus. I want to be clear that I find the study interesting and believe it absolutely deserves to be published to highlight the importance of these five sanctuaries as key sites for avifauna along the southeast coast of India. However, I think making an effort to identify another potential journal with a more descriptive focus could be a very good alternative.

---

## Round 0.3 · Major Revisions

After reviewing this revised version of your manuscript, I see that the main comments suggested by the reviewers have been included. However, there are still some details that need to be clarified before having a final version that can be published. Comments made by the reviewer are marked within the text of the manuscript.

Reviewer 1 ·

Basic reporting

This version is better but still requires significant improvement.

Experimental design

Some objectives need to be revised.
The methodology section should be revised.

Validity of the findings

The study can present particular value.
The results have to be revised.
The discussion should be revised as well.

Additional comments

I would revise the entire article and will structure it with sub-headings (if journal permits) to avoid duplication of the information and proper distribution of the information between sections.

Annotated reviews are not available for download in order to protect the identity of reviewers who chose to remain anonymous.

---

## Round 0.4 · Major Revisions

After reviewing this revised version of your manuscript, I see that the main comments suggested by the reviewers have been included. However, there are still some other details that need to be clarified before having a final version that can be published.

It is necessary that comments on the objectives, methodology, results and discussion are reviewed.

---

## Round 0.5 · accepted · Accept

After reviewing this revised version of your manuscript, I see that the main comments suggested by the reviewers have been included. Therefore, I am satisfied with the current version and consider it ready for publication.